# Non-invasive and noise-robust light focusing using confocal wavefront shaping

Dror Aizik [1] ✉ & Anat Levin[1]

Wavefront-shaping is a promising approach for imaging fluorescent targets deep inside scattering tissue despite strong aberrations. It enables focusing an incoming illumination into a single spot inside tissue, as well as correcting the outgoing light scattered from the tissue. Previously, wavefront shaping modulations have been successively estimated using feedback from strong fluorescent beads, which have been manually added to a sample. However, such algorithms do not generalize to neurons whose emission is orders of magnitude weaker. We suggest a wavefront shaping approach that works with a confocal modulation of both the illumination and imaging arms. Since the aberrations are corrected in the optics before the detector, the low photon budget is directed into a single sensor spot and detected with high signal-noise ratio. We derive a score function for modulation evaluation from mathematical principles, and successfully use it to image fluorescence neurons, despite scattering through thick tissue.

Scattering forms one of the hardest barriers on light-based approaches for tissue imaging. A promising way to overcome scattering, is a wavefront-shaping correction of the optical path. By using a spatial light modulator (SLM) device, one can reshape the coherent wavefront illuminating the sample, such that its aberration is conjugate to the aberration that will happen inside the tissue. When such a wavefront propagates through the sample, all incoming light can be focused into a small spot. In the same way, one can modulate the outgoing wavefront so that light photons emerging from a diffraction-limited target point are brought into a single sensor point, despite the tissue aberration. Unlike in ballistic-filtering approaches where scattered light is rejected, the main advantage of wavefront-shaping is that all light photons are used.

Earlier adaptive optics approaches used such modulations to correct aberrations in the optical path[1-3]. More recently, wavefront shaping techniques[4-6] have shown that it is possible to focus light through thick, highly-scattering samples[7-10].

Despite the large potential of the idea, finding the desired shape of the modulation correction is rather challenging. The desired modulation varies between different tissue samples and even varies spatially between different positions of the same tissue. For thick tissue, the modulation is a complex pattern containing a large number of free modes.

Earlier proof-of-concept demonstrations have used a validation camera behind the tissue to provide feedback to the algorithm[7-9,11-13], and other approaches have relied on the existence of a guiding star[4,10,14-22]. In the absence of such a guiding star, and when only non-invasive feedback is available, determining whether a wavefront has focused inside the tissue is not straightforward. The difficulty results from the fact that even if we can find an illumination wavefront that actually focuses into a small spot inside the tissue, the light back-scattering from this spot is aberrated again on its way to the camera, forming yet another scattered pattern.

Earlier approaches evaluate whether a wavefront modulation has focused using a multi-photon fluorescence feedback. In this way, the light emitted from a fluorescence spot is a non-linear function of the excitation intensity arriving to it, so when all light is focused into a single spot the total emission energy is maximized[2,15]. However, obtaining feedback using single-photon fluorescence is highly desired as the process is significantly simpler and cheaper than the multi-photon one. The single-photon case cannot be evaluated using the simple score function applied in the multi-photon case, since the emission energy is a linear function of the excitation energy and thus the amount of emission energy does not increase when all excitation power is focused in a spot. Recently, progress has been made on non-invasive wavefront shaping using single-photon feedback[23,24]. First,

[1]Department of Electrical and Computer Engineering, Technion, Haifa, Israel. ✉e-mail: droraizik@campus.technion.ac.il

Boniface et al.[23] have suggested that one can evaluate whether an incoming wavefront modulation has focused by computing the variance of the emitted speckle pattern. More recently, Aizik et al.[24] have suggested a rapid approach that can find a wavefront shaping modulation using iterative phase conjugation. Both approaches were only demonstrated when the fluorescent feedback was provided by synthetic fluorescent beads, which emit a relatively strong signal. In this work we aim to estimate the wavefront shaping modulations using feedback from fluorescent neurons rather than beads. These biological fluorescent samples impose two main differences. First, the targets are not sparse, but have wide continuous volumes. Thus, an initial excitation pattern is likely to produce a smooth image of emitted light rather than a speckle pattern, as illustrated in Fig. 1. Without speckle variation, the phase-retrieval process of[24] cannot be carried out. A second, more significant difference is the fact that the signal emitted from such samples is orders of magnitude weaker than the one provided by fluorescent beads. Due to bleaching, one can not increase the signal by increasing laser power or exposure time. Both algorithms[23,24] inherently assume that the speckle pattern emitted from a single fluorescent spot can be measured. However, if the number of fluorescent photons emitted from a neuron spot is low, and these photons are aberrated and spread over multiple sensor pixels, the number of photons captured by each pixel is very low. Thus, no speckle pattern can be observed and one can mostly measure noise, see visualization in

Fig. 2. An attempt to measure the variance of this image, as required by[23], will result in the noise variance rather than the speckle variance. While it is possible to improve the image in Fig. 2(c) by using a longer exposure or by increasing laser power, the optimization process of[23] requires thousands of shots, and bleaching happens well before its convergence.

This work proposes a confocal wavefront shaping framework, which can be applied in low-light scenarios and uses feedback from biological sources. To this end, we propose to use a simultaneous wavefront modulation both on the incoming excitation wavefront and on the outgoing emission[25,26], as illustrated in Fig. 1. The advantage is that since scattered photons are corrected in the optical path and we attempt to bring all photons emitted from a single spot into a single detector spot, we can measure them with a much higher signal-to-noise ratio (SNR).

To quantify the quality of a candidate modulation correction, we do not attempt to maximize the total energy emitted from the target. Rather, we seek to maximize the energy of the corrected wavefront *in a single spot*. We show that despite the fact that we use linear single-photon fluorescence, due to the double correction on both the illumination and imaging arms, our score function scales non-linearly with the intensity arriving at the fluorescence target. Thus, the returning energy at a single pixel is maximized by a focusing modulation that manages to bring all light into a single spot. We show that effectively,

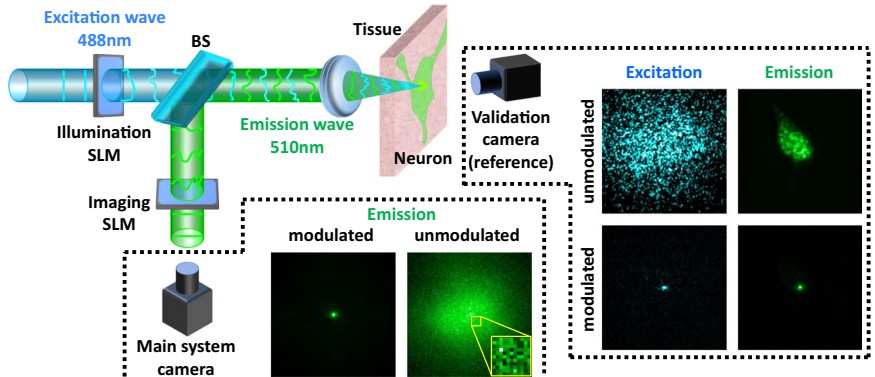

**Fig. 1 | System schematic.** An incoming laser illumination propagates through a layer of scattering tissue to excite a fluorescent neuron behind it. The emitted light is scattered again through the tissue toward a detector (main camera). In a conventional imaging system, the image of the neuron in the main camera is highly aberrated. Moreover, due to the weak emission, the image is very noisy. By adding two SLMs modulating the excitation and emission wavefronts, we can undo the tissue aberration. This allows us to reshape the incoming illumination into a spot on the neuron, and also to correct the emitted light and focus it into a spot in the main camera. For reference, we also place a validation camera behind the tissue, which can image the neuron directly and validate that the excitation light has focused into a spot. This camera does not provide any input to our algorithm. We visualize example images from these two cameras with and without the modulation correction.

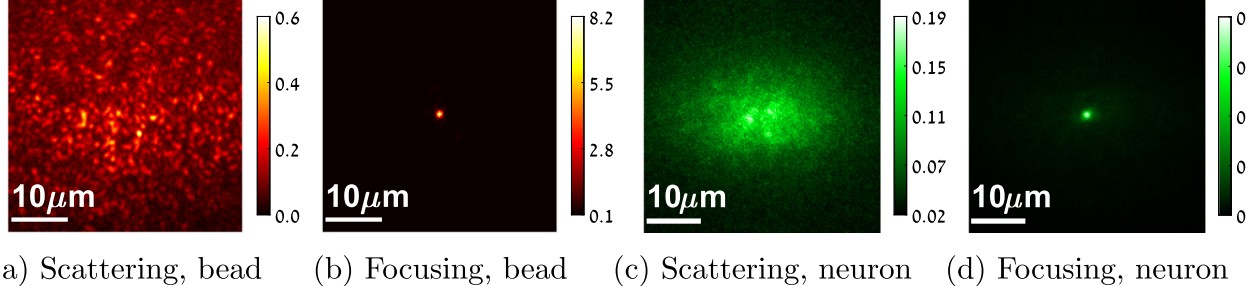

(a) Scattering, bead (b) Focusing, bead (c) Scattering, neuron (d) Focusing, neuron

**Fig. 2 | Types of fluorescent data. a, b** emission from invitrogen fluorescent microspheres (excitation/emission at 640/680 nm). A single bead is excited and the emitted light scatters through the tissue to generate a wide speckle pattern in (**a**). In (**b**) we use an aberration correction in the imaging arm so that the sensor measures a sharp spot. With such synthetic sources we can image a speckle pattern at high SNR, but this is not always the case with real biological samples. For example, (**c, d**) demonstrate fluorescent emission from EGFP neurons (excitation/emission at 488/ 508 nm), which is an order of magnitude weaker. In (**c**) a single fluorescent spot is excited, and the limited number of photons it emits are spread over multiple pixels. Noise is dominant, and an attempt to measure the variance of this image will evaluate the noise variance and not only the speckle variance. In (**d**) aberration correction is applied in the optics. As all photons are collected by a single pixel, SNR is drastically improved. Note that images (**c, d**) are taken under equal exposure and equal excitation power.

this score function is equivalent to the one used by previous two-photon fluorescence wavefront-shaping work[15].

We successfully use our algorithm to recover wavefront shaping modulations using fluorescent emission from EGFP (Enhanced Green Fluorescence Protein) neurons. By exploiting memory effect correlations, we use the modulation to locally image the shape of these neurons and their thin axons.

The following sections are organized as follows. We start by reviewing previous score functions for wavefront shaping modulations and understanding their restrictions. We then derive the confocal score and prove why it should correctly favor focusing modulations. Finally, we demonstrate experimental results where a confocal score is used to image neurons inside brain tissue.

## Results
### Imaging setup
Our wavefront shaping system follows the schematic of Fig. 1. A laser beam illuminates a tissue sample, and an SLM can modulate its shape. The illumination wavefront propagates through the scattering tissue and excites the fluorescent target behind it. We wish to image that target, but the emitted light is scattered again through the tissue on its way to the camera. A second phase SLM at the imaging arm can modulate the emitted light. Lastly, the modulated light is measured by the front main camera. In practice our SLMs are placed at the Fourier planes of the system, see methods section for a complete description.

The setup also includes a validation camera behind the tissue sample to assess focusing quality and capture a clean reference image of the same target. We emphasize that we only use non-invasive feedback by the main (front) camera, and the validation camera does not provide any input to the algorithm.

### Image formation model
We start with a mathematical definition of the images acquired by the above setup, which we will later use to formulate our optimization problem. Consider a set of $K$ fluorescent particles inside a sample, and denote their positions by $o_1, ..., o_K$. The illumination SLM displays a complex 2D field $\mathbf{u}$. We use $\mathbf{v}$ to denote a $K \times 1$ vector of the field propagating through the sample to each of the $K$ fluorescent sources. We can express $\mathbf{v} = \mathbf{T^i u}$, where $\mathbf{T^i}$ is the incoming transmission matrix, describing the forward coherent light propagation in the tissue. Likewise, the coherent propagation of light returning from the target to the SLM of the imaging arm can be described by a back-propagation transmission matrix $\mathbf{T^o}$.

Fluorescent energy emitted from a particle is proportional to $|\mathbf{v}_k|^{2\alpha}$, where $\alpha$ denotes the type of fluorescent excitation. The simplest case $\alpha = 1$ is known as single-photon fluorescence, where the emission is linear in the excitation energy $|\mathbf{v}_k|^2$. In two-photon fluorescence, $\alpha = 2$, and the emission is proportional to the squared excitation.

Since the laser energy is fixed, for any modulation $\mathbf{u}$ the energy arriving the fluorescent target is bounded and w.l.o.g. we assume

$$\sum_k |\mathbf{v}_k|^2 \le 1. \qquad (1)$$

### Scoring modulations
To estimate a wavefront shaping modulation, we first need a score function that can evaluate the focusing quality facilitated by a candidate modulation pattern, using a noninvasive feedback alone. We start by reviewing scores that were previously introduced in the literature and then explain our noise-robust confocal score.

Modulation evaluation is a simpler task when the same modulation can correct a sufficiently large isoplanatic image region. This assumption was made by adaptive optics research[1–3,27,28] and also by wavefront shaping approaches[29–32], who evaluate the quality of the

resulting image, in terms of contrast[2], sharpness, variance[29], or with a neural network regularization[32,33]. However, for thick tissue, wavefront shaping correction can vary quickly between nearby pixels, and a modulation may only explain a very local region. This case makes the above image quality scores less applicable, as inherently they evaluate the quality of an image region rather than a pixel. For spatially varying modulations, ideally, we need to evaluate the success of the modulation based on a per-pixel criterion. In the next paragraphs we review two previous local scores and understand their challenges.

### The total intensity score
Consider a configuration that only corrects the illumination arm, and the SLM in the imaging arm of Fig. 1 is not used. The simplest score that was considered in the literature[2,15] is just the total intensity (TI) measured over the entire sensor plane, which is also proportional to the total fluorescent power, reducing to

$$\mathcal{S}_{\text{TI}}(\mathbf{u}) = \sum_k |\mathbf{v}_k|^{2\alpha}. \qquad (2)$$

Since the energy at the target is bounded, as in Eq. (1), for the case $\alpha > 1$ this score is maximized when $\mathbf{v}$ is a one-hot vector, which equals 1 at a single entry and zero at all the others.

Two-photon fluorescence is expensive and hard to implement, and solutions that can use a single-photon excitation feedback are highly desirable. However, in the single-photon case where $\alpha = 1$, Eq. (2) reduces to the total power in $\mathbf{v}$, $\mathcal{S}_{\text{TI}}(\mathbf{u}) = \sum_k |\mathbf{v}_k|^2$, and since this power is fixed, the same amount of energy returns whether we spread the excitation power over multiple fluorescence sources or bring all of it into one spot.

### The variance maximization score
Boniface et al.[23] propose a score function for focus evaluation using a linear, single-photon feedback. The approach modulates only the illumination wavefront, attempting to focus light at a single spot, while the emitted light is scattered to the sensor. To score modulations, they measure the variance of the resulting speckle image and attempt to maximize it. The idea is that if we manage to focus all the excitation light at a single spot, the emitted light scattered through the tissue will generate a highly varying speckle pattern on the sensor plane. If the excitation is not focused, multiple sources emit simultaneously. The light emitted by these sources sums incoherently, and hence the variance of the speckle pattern on the sensor decays. They show that as in the two-photon case, speckle variance is proportional to $\sum_k |\mathbf{v}_k|^4$. Thus, as before, the speckle variance is maximized when the modulation focuses all the light into one spot and $\mathbf{v}$ is a one hot-vector.

The score was successfully demonstrated using feedback from fluorescent beads which were manually added to the sample. The emission from such beads is usually stronger than the one emitted from biological sources. Evaluating the variance score using the low SNR emission of biological sources is challenging. When a low number of photons is spread over multiple sensor pixels, the captured image is very noisy and an attempt to evaluate its variance will result in the noise variance rather than the speckle variance, see Fig. 2.

While one can reduce imaging noise by using a longer exposure or by increasing the power of the excitation laser, the optimization of[23] requires capturing many images of the same target, and usually the neuron bleaches well before convergence.

### The confocal score
In this work we evaluate a wavefront shaping modulation using a confocal score. While the two-photon and variance maximization scores corrected only the illumination arm, inspired by[25,26,34], we suggest to correct both arms. It has between previously noted that the image formation of a confocal single-photon microscope is equivalent

to that of a scanning two-photon microscope[35,36]. Hence, as in the two-photon case, a single-photon confocal measurment is a non linear function of flourescent excitation and can provide a discriminative score on candidate modulations.

Moreover, the fact that the SLM correction is applied in the optical path helps collecting all photons at one sensor pixel. For weak emitters this can drastically boost SNR.

We denote the modulations of the illumination and imaging arms by $\mathbf{u^i}, \mathbf{u^o}$. To score the focusing quality of each modulation we will use the intensity at the central pixel, rather than the total intensity throughout the sensor. In section 1 of the supplementary we prove that we can express the energy of the central pixel as:

$$\mathcal{S}_{\text{Conf}}(\mathbf{u^i}, \mathbf{u^o}) = \sum_k |v_k^o|^2 \cdot |v_k^i|^2, \qquad (3)$$

with $\mathbf{v^i} = \mathbf{T^i u^i}$ and $\mathbf{v^o} = \mathbf{u^o}^T \mathbf{T^o}$.

As mentioned in Eq. (1), the energy of $\mathbf{v^i}$ is bounded, and due to reciprocity the same applies for $\mathbf{v^o}$. It can be shown that this score is maximized when $\mathbf{v^i}, \mathbf{v^o}$ are both one-hot vectors, which bring all energy to a single joint entry $k$ and have zero energy at all other entries. That is, the score of Eq. (3) is maximized when the excitation modulation $\mathbf{u^i}$ brings all light to one of the particles $o_k$, and the modulation at the imaging arm $\mathbf{u^o}$ corrects the wavefront emitted from the same particle $o_k$ and brings all of it into the central pixel.

If the excitation and emission wavelengths are sufficiently similar, we explain in section 4 of the supplementary that it is best to use the same modulation at both the illumination and imaging arms, and the score of Eq. (3) reduces to $\sum_k |v_k|^4$ as in the variance-maximization and two-photon cases.

While the confocal score is equivalent to the variance maximization score above, it is significantly less susceptible to noise. This is due to the fact that the small number of photons we have at hand are collected at a single spot, rather than being spread over multiple pixels. Figure 2(c–d) shows the images emitted from a single neural spot with and without modulation in the imaging arm, and the significant noise reduction.

In this work we have explicitly optimized the confocal score of Eq. (3) using standard Hadamard basis approach[12], detailed in Supplementary section 2. Overall this optimization approach is significantly slower than the iterative phase conjugation of[24], but can handle dense targets. In our case the excitation and emission wavelengths are similar, yet not identical. While using two different modulations at the excitation and emission arms results in a modestly improved correction, it also doubles the required number of measurements. Alternatively, we can neglect the wavelength difference and use the same correction in both arms. Despite the approximation, the faster optimization is advantageous in the presence of photobleaching. We compare a single correction to two corrections in supplementary section 4.

In section 5 of the supplementary file we compare our confocal score with the variance maximization approach of[23], showing that our approach can converge using a significantly smaller number of photons. We also compare against one of the non-local approaches by[29]. This approach assumes that a single modulation can correct a wide image region rather than a single spot. Our evaluation shows that when memory-effect (ME) exists over a wide extent, this algorithm can indeed recover good modulations, but the quality of the results degrades for a small ME, where the size of the iso-planatic patches that can be corrected with a single modulation is small.

## Experimental results

We image slices of mice brain with EGFP neurons, excited at 488 $nm$ and imaged at 508 $nm$. We use two types of aberrations. In the first case we use thin brain slices of thickness 50 $\mu m$, which are almost aberration-free. We generate scattering by placing these slices behind

a layer of chicken breast tissue (200 – 300 $\mu m$ thick), or parafilm whose optical properties were measured in[23]. The advantage is that since the fluorescence is present only in a thin 2D layer we can obtain a clean reference from a validation camera. In a second experiment we image inside thick brain slices. The slice were originally cut to be 400 $\mu m$ thick, though while squeezing between two cover-glass some of the water evacuated and the resulting slice is somewhat thinner. Since the fluorescent components are spread in 3D it is not always possible to capture clear aberration-free references. More details about the mice are included in the methods section.

In Fig. 3 we visualize some results of our algorithm. Figure 3(a) shows an image of the initial excitation pattern from the validation camera behind the tissue. As can be observed, the tissue exhibits significant scattering. In Fig. 3(b), we visualize the excitation light after optimizing the wavefront shaping modulation, which is nicely focused into a sharp spot. In Fig. 3(c–d), we also show the emitted light. Before optimization a wide area is excited and we can see the neuron shape. At the end of the optimization a single point is excited. In Fig. 3(e–f), we visualize the views of the front main camera, providing the actual input to our algorithm. Before optimization the emitted light is scattered over a wide sensor area. As a low number of photons is spread over multiple sensor pixels, the captured image is noisy. At the end of the optimization the aberration is corrected and all the photons are brought into a single sensor pixel. As indicated by the colorbar, the spot at the focused images received a much higher number of photons compared to the wide scattering images of unfocused light. This holds despite the fact that all images were captured under equal exposure and equal excitation power. In Fig. 3(g), we demonstrate the actual point spread function of the tissue aberration. For that, after convergence we keep the modulation in the illumination arm so that we excite a single spot, but we replaced the modulation on the imaging SLM with a blank one, so the emitted light is not corrected. One can see that the aberration of a single fluorescent spot is rather wide.

To better appreciate the noise handled by our algorithm, in Fig. 4 we visualize a 41 × 41-pixel window captured by the main camera, at a few iterations of our algorithm. In the beginning this image is very noisy, because a small number of emitted photons are spread over multiple sensor pixels. However, when the optimization proceeds, it finds a better modulation correction. As a result, all the laser power is brought to excite one spot and all the emitted photons are collected to one sensor spot, thus they are measured with better SNR.

We use the recovered wavefront shaping modulation to image a wide area rather than a single spot. In Fig. 5 we demonstrate results for a thin brain slice beyond chicken breast and parafilm. For that, we excite a wide area and use a correction only at the imaging arm. Due to the memory effect[37,38], the same modulation can allow us to image a small local patch rather than a single spot. With the correction, the neuron is observed with a much higher contrast and even the axons (thin lines around the neuron) emerging from it, whose emission is much weaker, can be partially observed. Additional results are provided in Figs. 4 and 5 of the supplementary file.

We note that noise is less visible in the aberrated images of Fig. 5(a), because these images are captured with a much longer exposure compared to the optimization images in Fig. 4. While it is possible to capture a few noise-free images of such targets, it is not possible to do that without bleaching throughout the optimization. In Fig. 5 we mark with an arrow some points at which the algorithm has converged. One could see a darker spot as such points have bleached during optimization.

In Fig. 6 we show imaging through a 400 $\mu m$ brain slice. Due to the 3D structure of the target, to isolate a neuron at a single-depth plane we have to use a slow confocal scanning, where the modulation is placed on both arms and is tilt-shifted to excite and image different spots of the target. We compare this with an uncorrected confocal scan that is significantly aberrated. In some cases there is a small shift between the corrected and uncorrected confocal images, because the

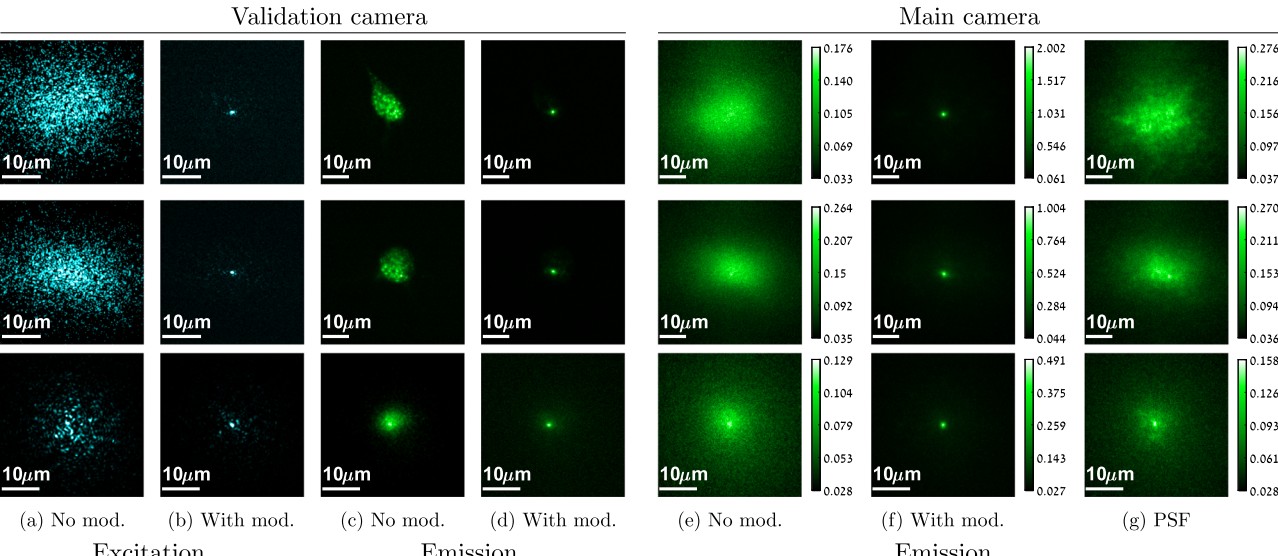

(a) No mod.  (b) With mod.  (c) No mod.  (d) With mod.  (e) No mod.  (f) With mod.  (g) PSF

Excitation        Emission                      Emission

**Fig. 3 | Wavefront shaping results.** We visualize views from the validation and main cameras, each row demonstrates a different tissue sample. **a-b** The excitation light as viewed by the validation camera at the back of the tissue. Due to significant scattering, at the beginning of the algorithm when no modulation (mod.) is available, a wide speckle pattern is generated. After optimization, the modulated wavefront is brought into a single spot. **c-d** By placing a band-pass filter on the validation camera, we visualize the emitted light with and without the modulation correction. **e-f** Views of the emitted light at the main front camera with and without the modulation correction. Note that this is the only input used by our algorithm. Without modulation, light is scattered over a wide image area and the image is noisy. A sharp clean spot can be imaged when the limited number of photons are brought into a single sensor pixel. **g** By correcting the emission such that a single spot is excited and leaving the imaging path uncorrected, we can visualize the actual aberration of a single fluorescent point source. The top two examples used a thin brain layer behind parafilm, and the lower one is a thick brain slice.

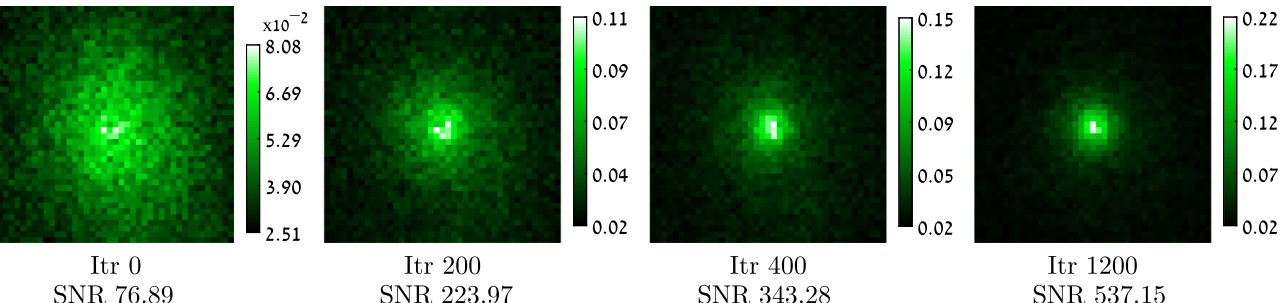

Itr 0          Itr 200        Itr 400        Itr 1200
SNR 76.89      SNR 223.97     SNR 343.28     SNR 537.15

**Fig. 4 | Convergence with noise.** We visualize images captured by the main camera at a few iterations of our algorithm. In the beginning a small number of photons are spread over multiple sensor pixels and the resulting image is very noisy. As the algorithm proceeds and a wavefront shaping modulation is recovered, the low number of photons is brought to a single sensor spot and the measured image has a higher SNR. SNR is calculated by capturing 40 images with the same modulation and evaluating their variance.

recovered modulation has also shifted the focal spot inside the target. Note that our confocal scanning is currently implemented by tilting and shifting the modulation pattern on the SLM and not with a proper galvo mirror. Since this approach is very slow we could only scan small windows. We also include a full frame image from the validation camera behind the tissue, but due to the 3D fluorescence structure, in some cases this does not provide a clear ground truth. Additional results are provided in Fig. 6 of the supplementary file.

## Discussion

In this research we have analyzed score functions for wavefront shaping correction, using non-invasive feedback, at the absence of a guiding star. To assess focusing quality, we seek a score function that can measure a non-linear function of the light emitted by different sources. This is naturally achieved when using two-photon fluorescent feedback, but is harder to achieve with linear fluorescence. We show that by using a confocal correction at both the illumination and imaging arms we can measure such a non-linear feedback, which is maximized when all excitation light is brought into one spot.

Moreover, the fact that our system uses a correction of the emitted light as part of the optical path allows us to bring the limited number of emitted photons into a single sensor spot, facilitating a high SNR measurement.

As analyzed by[15], a two-photon feedback may not favor a focusing modulation if the fluorescent target occupies an unbounded three dimensional volume, and convergence happens because most fluorescent targets of interest are sparse. Our confocal approach uses a similar score and should suffer from the same limitation.

While our approach can recover focusing modulations, it can converge to different spots on the fluorescent target, depending on initialization. Despite the fact that we cannot know where it has focused, we can use memory effect correlations to image the surrounding window.

Another drawback of the approach is that in our current implementation it takes about 30 min to optimize for one modulation pattern. Some of this can be largely optimized with better hardware, such as a faster SLM. However, the iterative optimization is inherently slower than the power iterations of[24]. Beyond better hardware, we are exploring algorithmic alternatives which can accelerate optimization.

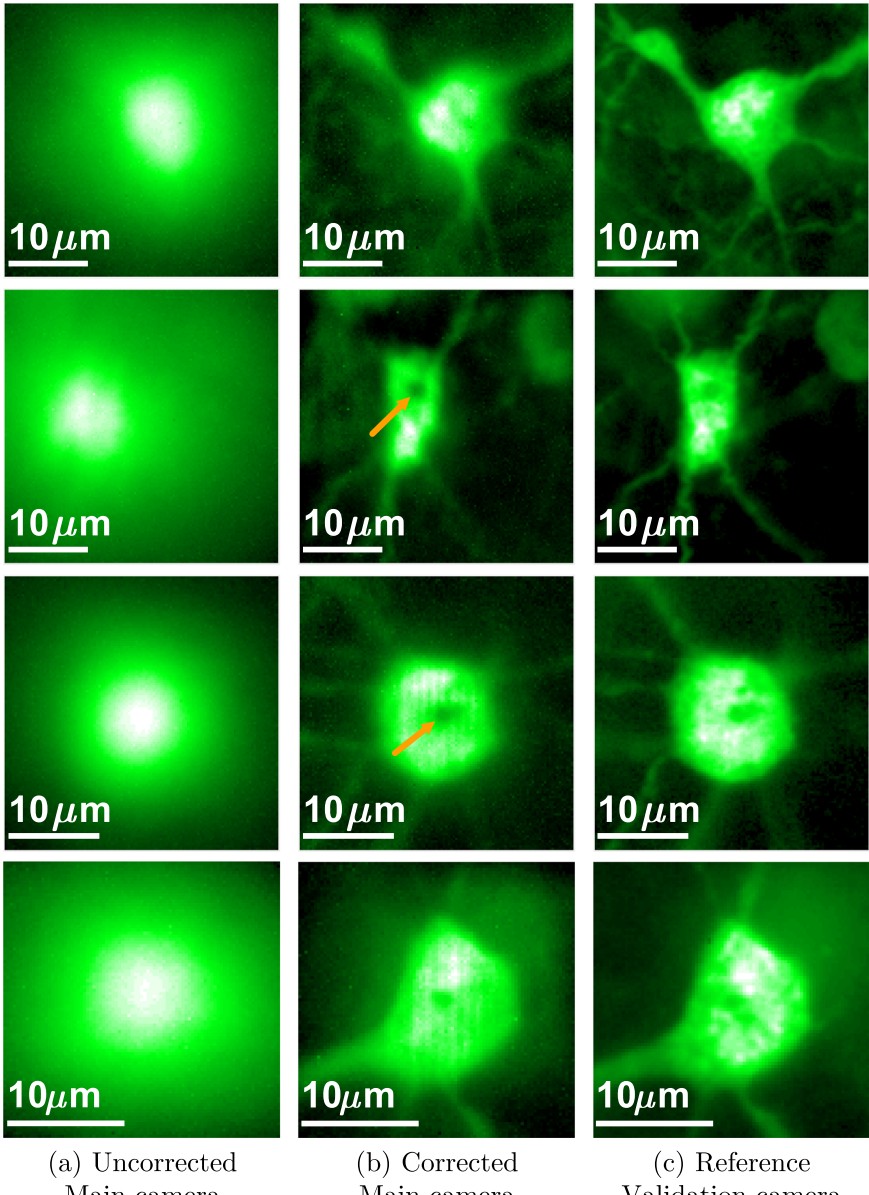

(a) Uncorrected
Main camera

(b) Corrected
Main camera

(c) Reference
Validation camera

**Fig. 5 | Imaging a wide area.** We image a thin fluorescent brain slice behind a scattering layer. The top two images correct scattering through chicken breast tissue and the lower ones through parafilm. **a** Image of the neuron from the main camera with no correction, strong scattering is present and the neuron structure is lost. **b** Image with our modulation correction, the neuron shape as well as some of the axons are revealed. **c** A reference image of the same neuron, from the validation camera. The arrow marks a spot at which the optimization has converged. This spot is darker as it bleached during optimization.

## Methods

### System design

In Fig. 2 of the supplementary material we visualize the full imaging setup for our wavefront-shaping correction. All components are listed in Table 1 of the supplementary material. A laser beam illuminates a tissue sample via a microscope objective. A phase SLM in the illumination arm modulates the illumination pattern. The illumination wavefront propagates through the scattering tissue and excites the fluorescent target behind it. We wish to image that target, but the emitted light is scattered again through the tissue on its way to the objective. Scattered light is collected via the same objective, and reflected at a dichroic beam-splitter. A second phase SLM at the imaging arm modulates the emitted light. Lastly, the modulated light is measured by the front main camera. In our setup the SLMs are placed in the Fourier plane of the system. We use a $10\,nm$ bandpass filter in the imaging arm to image a relatively monochromatic light. As we want to correct the scattering of the sample itself rather then aberrations in the

optical path, before starting the optimization we place the sample so the fluorescent light has smallest support in the main camera, then focus the objective of the validation camera (Obj 2 in Fig. 2 of the supplementary material) such that the neuron is in focus. We elaborate on the calibration and alignment of this setup in section 7 of the supplementary material.

### Experimental targets

All experiments were approved by Institutional Animal Care and Use Committee (IACUC) at the Technion (IL-149-10-2021), as well as the Hebrew University of Jerusalem (MD-20-16065-4). We image slices of mice brain with EGFP neurons, excited at $488\,nm$ and imaged at $508\,nm$. We used two types of aberrations. In the first case we used thin, brain slices of thickness $50\,\mu m$ which are almost aberration free, and generate scattering by placing these slices behind a layer of chicken breast tissue ($200-300\,\mu m$ thick) or parafilm whose optical properties were measured in[23]. In a second experiment we image

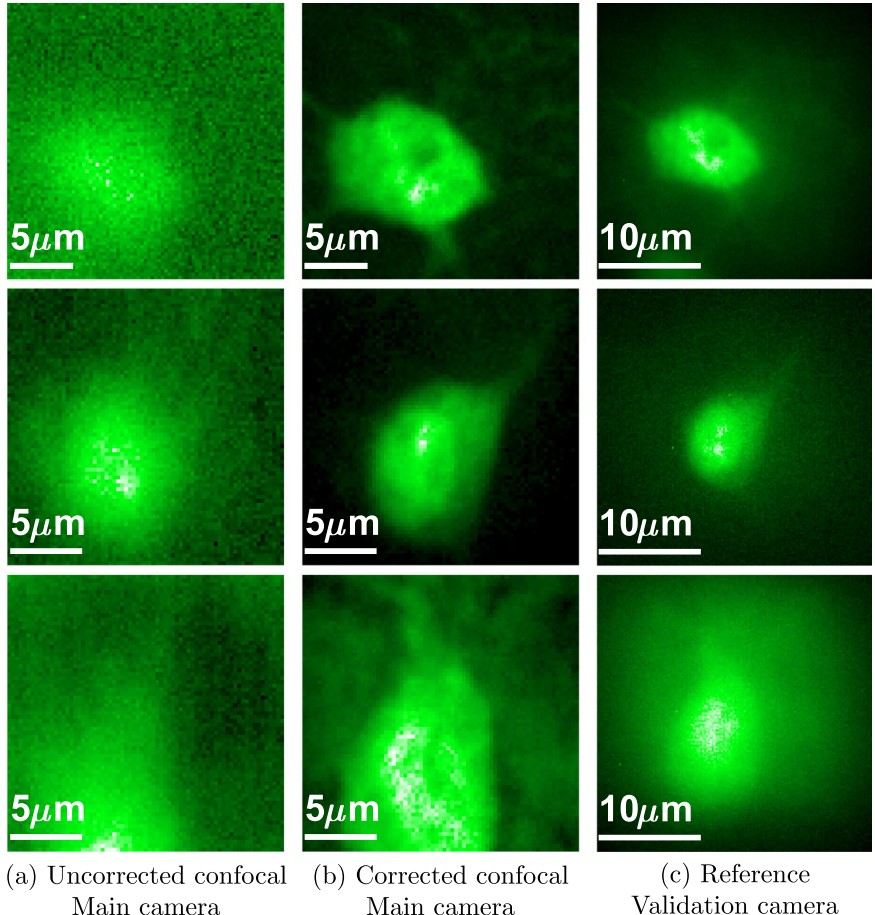

(a) Uncorrected confocal
Main camera

(b) Corrected confocal
Main camera

(c) Reference
Validation camera

**Fig. 6 | Imaging inside a thick fluorescent brain slice.** We image a wide 3D target inside a 400 $\mu m$ thick fluorescent brain slice. **a** A confocal image of the neuron from the main camera with no correction, strong scattering is present and the neuron structure is lost. **b** A confocal image with our modulation correction, the neuron shape as well as some of the axons are revealed. **c** A reference image of the same neuron, from the validation camera. Due to the 3D spreading of the fluorescent components, the validation camera cannot always capture an aberration-free image of the target.

through thick brain slices. The slice were originally cut to be 400 $\mu m$ thick, though while squeezing between two cover-glass some of the water evacuated and the resulting slice is somewhat thinner. The thin slices contain Betz neurons coming from a 6 w female Strain:$C57BL6$ mouse. The thick slices contain pyramidal cells of layer 2-3 in the cortex, from a triple transgenic mouse, $Rasgrf2 - 2A - dCre$; $CamK2a - tTA$;$TITL - GCaMP6f$ line 7 m old.

## Reporting summary

Further information on research design is available in the Nature Portfolio Reporting Summary linked to this article.

## Data availability

The data that support the findings of this study is available in ref. 39.

## Code availability

The code used for acquiring and processing the data is available at ref. 39.

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

## Acknowledgements

We thank the following people in their help preparing neural samples: Amit Parizat, Etay Aloni, Zhige Lin and Amit Zeisel from the Technion department of Biotechnology and Food Engineering, as well as Ariel Gilad and Odeya Marmor-Levin from the Medical Neurobiology department at the Hebrew University. AL thanks the European Research Council [101043471] for funding this research.

## Author contributions

Both authors contributed extensively to the work presented in this paper.

## Competing interests

The authors declare no competing interests.
