## [Peer Review File · Nature Communications]

Non-invasive and noise-robust light focusing using confocal wavefront shapingREVIEWER COMMENTS

Reviewer #1 (Remarks to the Author):

The paper describes a form of adaptive optics for correcting phase aberrations introduced in fluorescence microscopes. Specifically, they authors seek to overcome some of the limitations in existing methods by performing “confocal modulation” of the illumination and imaging arms. Many other demonstrations have relied upon nonlinear fluorescence – particularly two-photon – which makes the optimization process easier. Other recent work, appropriately cited in this paper, has used speckle variance as the optimized quantity. The authors proposed to use a form of confocal filtering as an alternative approach. A distinction in the present paper is the use of two spatial light modulators, one in the illumination path and one in the detection path. The motivation is clear and the methods and results are well explained. The method clearly works.

While the work has been well executed and is well-presented, I am of the opinion that it is not a significant and high impact advance. The idea of “confocal correction” is not fundamentally new, as it is the basis of previous adaptive correction in any confocal laser scanning microscope (as they have correctly cited in their reference 25 from over twenty years ago). Previous implementations in such microscopes have used a single correction adaptive device to correct both illumination and imaging paths on the assumption that the path length aberration is the same in both paths. This assumption is generally reliable for low-order smooth aberrations, but might become invalid if the scattering is significant. This is where using two separate correction devices might be necessary. However, there is no evidence presented in the paper that the correction provided by the two modulators is significantly different. If not, then there is no benefit in this method over previous forms of signal optimization using a single correction device. The only correction patterns presented are in supplementary figure 6, which shows tilt-memory effect. The imaging results shown for example in figure 5 show fairly good correction over a field of view of approximately 40 micrometers, which indicates good anisoplanatism (or field independent aberrations, or wide memory effect, dependent on chosen phrasing). This hints that there may not be significant variations in correction phase between the paths, although it is difficult to know without seeing the phase directly.

In conclusion, this work is novel and a useful contribution to the field. However, there is not a clearly useful advance over existing methods and the demonstrations of correction through semi-artificial specimens, in that the aberrations are introduced by chicken breast tissue and parafilm. The work is hence not suited to this journal, but should certainly be published elsewhere.

Reviewer #2 (Remarks to the Author):

In this manuscript, the authors present a new adaptive optics algorithm based on maximizing one-photon fluorescence in a microscope under confocal configuration. The authors have adeptly summarized the current advancements in the adaptive optics (AO) field and clearly explained the motivation behind their work. Their objective was to develop an image-guided, rather than guide-star-based, adaptive optics algorithm that operates without relying on large isoplanatic patches or multiphoton fluorescence signal, and can effectively converge with few photons. I am impressed by the ingenious solution they have conceived, and I believe this manuscript is fitting for publication in Nature Communications.

Before delving into my technical comments, I'd like to highlight the idea of this study. Any AO algorithms that function by maximizing a metric requires the metric to improve with a focus tighter and closer to being diffraction-limited. There are primarily three methods to yield such a metric:

1. Employ multiphoton excitation whose intensity grows steeply with peak intensity.
2. Computationally determine a nonlinear metric derived from the images, such as variation or nonnegative log loss.

3. Utilize the confocal configuration to introduce nonlinearity, which is the innovation in this paper.

The advantage of approach (3) is that it concentrates light on a single pixel of detector, potentially enabling a high signal-to-noise ratio for swift optimization, without necessitating extensive scanning of the focal plane.

While I'm convinced of the value this research can add to the AO field, I'd appreciate further clarity on certain aspects of this novel AO approach:

1. AO algorithms using two-photon fluorescence aren't always guaranteed to converge, even if they frequently do in practical experiments. This stems from the fact that the integral of a two-photon signal in a uniformly stained, infinite volume remains a constant, irrespective of the focus size. Essentially, a tighter focus might not yield a stronger signal. Real-world stained objects, being finite, typically ensure that a sharper focus enhances the signal. Yet, occasionally, this property might result in a diminishing gradient, causing sluggish AO algorithm convergence. Given that the confocal detection point spread function is quite similar to two-photon (being intensity squared), were there instances where your algorithm faced difficulties in convergence? Could the authors share some examples and elucidate the issues impeding convergence?

2. The proposed scheme necessitates a second SLM to focus light on the main camera, which inevitably complicates the system and might influence its speed. Could the authors elucidate the feasibility of employing a single SLM for the experiment to process both excitation and emission wavelength, especially since the phase patterns on both SLMs are reportedly identical?

3. A related question is: if two SLMs were used, and the correction patterns were to differ on the two SLMs, how would you modify the optimization algorithm?

4. A discernible limitation in this study is the exclusive demonstration on a 50 μm brain slice. There isn't sufficient experimental data to convince readers how the performance of this new method compares to existing techniques, or its applicability in important applications like deeper tissue imaging or in vivo imaging. Although I recognize the proof-of-concept nature of this research, I'd like to gauge how this new method fares against established techniques, such as those in Ref. 23. and the following work:

Yeminy, Tomer, and Ori Katz. "Guidestar-free image-guided wavefront shaping." *Science advances* 7.21 (2021): eabf5364.

5. Despite the neuron images shown in the paper, I wish to have more data to examine how AO correction falls off toward the edge of a larger field-of-view after a single-point wavefront correction in the middle of the FOV.

6. I'm keen to hear the authors' perspective on implementing their methodology in imaging modalities beyond one-photon imaging, such as multiphoton and OCT.

In conclusion, I wish to reiterate my enthusiasm for this study and endorse its publication. I eagerly await a refined version addressing my queries.

Reviewer #3 (Remarks to the Author):

The authors introduce a novel method to optimize linear fluorescence images in the presence of strong scattering. By optimizing both the illumination and detection arms, and focusing on maximizing the central pixels on the camera, the authors' method essentially replicates the function of a wavefront shaping system for a confocal design. The concept holds merit and should intrigue the wavefront shaping community. However, there are several aspects that need further clarification for a conclusive decision:

1. Signal-to-Noise Ratio (SNR): The authors highlight their ability to overcome the SNR limitations

compared to prior works. However, they do not provide a quantification of the SNR in their images. I recommend adding this analysis to substantiate their claim.

2. Comparison with Variance Optimization: The authors mention that their end metric is largely analogous to variance optimization, yet claim superior performance with scattering tissue samples. Without experimental evidence, this assertion remains dubious. Providing this evidence would significantly bolster the integrity of their work.

3. SLM Correction Patterns: There's an absence of correction patterns from the SLMs for both arms. It would enhance clarity and comprehension if these patterns were presented alongside the results.

4. Details and Compliance: The manuscript omits several critical details, including specifications about the SLMs, filters, and dichroics used in the setup. Particularlry, is the filter narrow band so the authors can assume it's close to coherent case? Additionally, it lacks details about the age and source of the mouse brain sample, along with the necessary regulatory compliance statements.

5. Emission Path Assumptions: The authors postulate modulation of the emission path, assuming some degree of coherency. Given the potential for broadband emission in real-life scenarios, how does the proposed metric fare when the emission path transitions to linear incoherence (where only intensity modulation is viable)?

6. Citations: I suggest referencing the work of Sinefeld, David, et al. titled "Three-photon adaptive optics for mouse brain imaging." (Frontiers in neuroscience 16, 2022: 880859). This paper demonstrates the feasibility of adaptive optics (wavefront shaping) for fluorescence imaging with thin fluorescence samples, which provides intuition and explanation that performing 1-photon fluorescence wavefront shaping may not be challenging for thin brain slice in the author's case.

7. Optimization Details and Results Discussion: The main text is rather scant on optimization specifics and a thorough discussion of the primary findings. I suggest the authors to add some essential information about these to the main text.

Minor Comments:

1. Figures & Colorbars: Neuronal figures lack colorbars, which makes it appear that the uncorrected images might be saturated. It would be beneficial to include colorbars without normalization to depict the actual values.

2. Uncited Reference: Reference [35] is conspicuously uncited in the manuscript.

3. Terminology Clarification: In the main text, line 245, the term "one-hot vectors" is used. Could the authors clarify if they mean that these vectors are binary?

4. Typographical Error: There's a typo in the main text, line 179. "then" should be corrected to "than". Please carefully proofread the manuscript.

Dear reviewers,

We thank you all for your time and for your thoughtful comments. We attach a revised manuscript, where we made our best attempt to address all comments and suggestions. The major additions are:

- 1) We have applied our technique to image through a 400um brain slice, rather than the previous results, which used a thin brain slice behind a different scattering layer.
- 2) We include a detailed comparison against two alternative algorithms: the variance maximization approach of Boniface et al, and the non-local correction of Yeminny et al.
- 3) We perform a careful comparison between using two different modulations for the emission and excitation wavefronts, and the approximation using the same modulation in both arms. Our observation is that two modulations facilitate a better correction, but also double the number of required measurements. Thus, in practice, in the presence of photo-bleaching, using the same modulation in both arms yields better results.
- 4) We add a measurement of the extent of the memory effect in our samples.

Please find below detailed responses to reviewing comments.

Reviewer #1 (Remarks to the Author):

The paper describes a form of adaptive optics for correcting phase aberrations introduced in fluorescence microscopes. Specifically, they authors seek to overcome some of the limitations in existing methods by performing “confocal modulation” of the illumination and imaging arms. Many other demonstrations have relied upon nonlinear fluorescence – particularly two-photon – which makes the optimization process easier. Other recent work, appropriately cited in this paper, has used speckle variance as the optimized quantity. The authors proposed to use a form of confocal filtering as an alternative approach. A distinction in the present paper is the use of two spatial light modulators, one in the illumination path and one in the detection path. The motivation is clear and the methods and results are well explained. The method clearly works.

While the work has been well executed and is well-presented, I am of the opinion that it is not a significant and high impact advance. The idea of “confocal correction” is not fundamentally new, as it is the basis of previous adaptive correction in any confocal laser scanning microscope (as they have correctly cited in their reference 25 from over twenty years ago). Previous implementations in such microscopes have used a single correction adaptive device to correct both illumination and imaging paths on the assumption that the path length aberration is the same in both paths. This assumption is generally reliable for low-order smooth aberrations, but might become invalid if the scattering is significant. This is where using two separate correction devices might be necessary. However, there is no evidence presented in the paper that the correction

provided by the two modulators is significantly different. If not, then there is no benefit in this method over previous forms of signal optimization using a single correction device. The only correction patterns presented are in supplementary figure 6, which shows tilt-memory effect. The imaging results shown for example in figure 5 show fairly good correction over a field of view of approximately 40 micrometers, which indicates good anisoplanatism (or field independent aberrations, or wide memory effect, dependent on chosen phrasing). This hints that there may not be significant variations in correction phase between the paths, although it is difficult to know without seeing the phase directly.

In conclusion, this work is novel and a useful contribution to the field. However, there is not a clearly useful advance over existing methods and the demonstrations of correction through semi-artificial specimens, in that the aberrations are introduced by chicken breast tissue and parafilm. The work is hence not suited to this journal, but should certainly be published elsewhere.

We now include images through a 400um thick brain slice, and not only thin brain slices behind chicken breast as in the original submission.

Also we have added in supplementary section 5 an exact comparison between the usage of two different modulations in each wavelength and the approximation of using the same modulation in both arms. Our observation is that two modulations facilitate a better correction, but also double the number of required measurements. Thus, in practice, in the presence of photo-bleaching, using the same modulation in both arms yields better results.

We have added images of the recovered modulations to supplementary figure 9.

Regardless of the above experiment, we do not view the usage of two different modulations as the main contribution of this paper. Rather, the two main contributions are:

1) A mathematical analysis of previously introduced scores which proves why they should favor a focusing modulation, and a proof that a confocal score results in a non-linear function, like a 2-photon score. To our best knowledge no derivation of this form was ever provided in the literature.

2) Previous experimental applications of a confocal correction were only demonstrated on very simple aberrations, while our work provides the first demonstration of this principle to remove severe tissue scattering.

Reviewer #2 (Remarks to the Author):

In this manuscript, the authors present a new adaptive optics algorithm based on maximizing one-photon fluorescence in a microscope under confocal configuration. The authors have adeptly summarized the current advancements in the adaptive optics (AO)

field and clearly explained the motivation behind their work. Their objective was to develop an image-guided, rather than guide-star-based, adaptive optics algorithm that operates without relying on large isoplanatic patches or multiphoton fluorescence signal, and can effectively converge with few photons. I am impressed by the ingenious solution they have conceived, and I believe this manuscript is fitting for publication in Nature Communications.

Before delving into my technical comments, I'd like to highlight the idea of this study. Any AO algorithms that function by maximizing a metric requires the metric to improve with a focus tighter and closer to being diffraction-limited. There are primarily three methods to yield such a metric:

1. Employ multiphoton excitation whose intensity grows steeply with peak intensity.
2. Computationally determine a nonlinear metric derived from the images, such as variation or nonnegative log loss.
3. Utilize the confocal configuration to introduce nonlinearity, which is the innovation in this paper.

The advantage of approach (3) is that it concentrates light on a single pixel of detector, potentially enabling a high signal-to-noise ratio for swift optimization, without necessitating extensive scanning of the focal plane.

While I'm convinced of the value this research can add to the AO field, I'd appreciate further clarity on certain aspects of this novel AO approach:

1. AO algorithms using two-photon fluorescence aren't always guaranteed to converge, even if they frequently do in practical experiments. This stems from the fact that the integral of a two-photon signal in a uniformly stained, infinite volume remains a constant, irrespective of the focus size. Essentially, a tighter focus might not yield a stronger signal. Real-world stained objects, being finite, typically ensure that a sharper focus enhances the signal. Yet, occasionally, this property might result in a diminishing gradient, causing sluggish AO algorithm convergence. Given that the confocal detection point spread function is quite similar to two-photon (being intensity squared), were there instances where your algorithm faced difficulties in convergence? Could the authors share some examples and elucidate the issues impeding convergence?

We agree that in theory, given an unbounded volume, our score may be invariant to focus, as in the two-photon case.

In practice, for any synthetic simulation we have run, the algorithm has always converged to the desired optimum, without any diminishing gradients. In the lab there are many convergence problems, but we attribute them to all sorts of experimental troubles such as bleaching and vibrations, rather than the actual landscape of the cost function.

We have included a failure example in supplementary figure 3.

2. The proposed scheme necessitates a second SLM to focus light on the main camera,

which inevitably complicates the system and might influence its speed. Could the authors elucidate the feasibility of employing a single SLM for the experiment to process both excitation and emission wavelength, especially since the phase patterns on both SLMs are reportedly identical?

We have now added in supplementary section 5 an exact comparison between the usage of two different modulations in each wavelength and the approximation of using the same modulation in both arms. Our observation is that two modulations facilitate a better correction, but also double the number of required measurements. Thus, in practice, in the presence of photo-bleaching, using the same modulation in both arms yields better results.

We agree that given this conclusion, one can also implement the system with a single SLM. Our original motivation for using two SLMs was: first, the option to use two different corrections, and second, for debugging purpose it is sometimes useful to be able to correct only one of the arms. However, these are our personal considerations, and it is possible that the decision to use two SLMs was not the ideal choice.

3. A related question is: if two SLMs were used, and the correction patterns were to differ on the two SLMs, how would you modify the optimization algorithm?

See supplementary sec 2.2. We alternate between modifying the illumination SLM and the imaging SLM.

4. A discernible limitation in this study is the exclusive demonstration on a 50 μm brain slice. There isn't sufficient experimental data to convince readers how the performance of this new method compares to existing techniques, or its applicability in important applications like deeper tissue imaging or in vivo imaging. Although I recognize the proof-of-concept nature of this research, I'd like to gauge how this new method fares against established techniques, such as those in Ref. 23. and the following work:

Yeminy, Tomer, and Ori Katz. "Guidestar-free image-guided wavefront shaping." *Science advances* 7.21 (2021): eabf5364.

We have added to supplementary file sec 6.2 a detailed comparison between our approach and that of Yeminy et al. Due to photo-bleaching, it is not possible to run two algorithms on the same sample under equal noise conditions. Rather we chose to run a comparison using synthetic transmission matrices. An advantage of the synthetic approach is that we can generate transmission matrices with different extents of iso-planatic patches. As expected our comparison shows that when the iso-planatic patch is wide both algorithms can recover the desired modulation, but as the extent of the memory effect decreases, our local confocal correction can recover sharper modulations.

5. Despite the neuron images shown in the paper, I wish to have more data to examine how AO correction falls off toward the edge of a larger field-of-view after a single-point wavefront correction in the middle of the FOV.

We added a measurement of the extent of the ME in the supplementary file, figure 7.

6. I'm keen to hear the authors' perspective on implementing their methodology in imaging modalities beyond one-photon imaging, such as multiphoton and OCT.

Our score function relies on the fact that the emission is incoherent. In a coherent setup such as OCT we have noticed that due to interference, it is possible to maximize the confocal score and arrive at a sharp spot on the sensor, even if the light does not focus into a single spot inside the tissue.

In conclusion, I wish to reiterate my enthusiasm for this study and endorse its publication. I eagerly await a refined version addressing my queries.

Reviewer #3 (Remarks to the Author):

The authors introduce a novel method to optimize linear fluorescence images in the presence of strong scattering. By optimizing both the illumination and detection arms, and focusing on maximizing the central pixels on the camera, the authors' method essentially replicates the function of a wavefront shaping system for a confocal design. The concept holds merit and should intrigue the wavefront shaping community. However, there are several aspects that need further clarification for a conclusive decision:

1. Signal-to-Noise Ratio (SNR): The authors highlight their ability to overcome the SNR limitations compared to prior works. However, they do not provide a quantification of the SNR in their images. I recommend adding this analysis to substantiate their claim.

We have now added SNR scores to the images in different iterations in figure 4.

2. Comparison with Variance Optimization: The authors mention that their end metric is largely analogous to variance optimization, yet claim superior performance with scattering tissue samples. Without experimental evidence, this assertion remains dubious. Providing this evidence would significantly bolster the integrity of their work.

We have added in supplementary Sec 6.1 a detailed comparison between our approach and the variance maximization approach. As mentioned, due to photo-bleaching it is not possible to run two algorithms on the same sample under equal noise conditions. Rather we chose to run a comparison using synthetic transmission matrices. An advantage of the synthetic approach is that we can test different noise levels and a different amount of scattering. As expected, our comparison shows that when the scattering is larger and the speckle pattern is wider, the SNR gain of our algorithm increases. While both algorithms can converge to a good modulation, our approach can do so using a significantly lower number of photons.

3.SLM Correction Patterns: There's an absence of correction patterns from the SLMs for both arms. It would enhance clarity and comprehension if these patterns were presented alongside the results.

These patterns typically look like noise patterns and do not add much intuition. However, for completeness, we have now added such patterns in supplementary figure 9.

4.Details and Compliance: The manuscript omits several critical details, including specifications about the SLMs, filters, and dichroics used in the setup. Particularlry, is the filter narrow band so the authors can assume it's close to coherent case? Additionally, it lacks details about the age and source of the mouse brain sample, along with the necessary regulatory compliance statements.

Yes, we use a 10um narrowband filter on the emitted light to increase its monochromaticity. We have added an exact list of components and details about the mice in supplementary sec 3. Permission numbers by the Institutional Animal Care and Use Committee are reported at 2.3 of the main paper.

5. Emission Path Assumptions: The authors postulate modulation of the emission path, assuming some degree of coherency. Given the potential for broadband emission in real-life scenarios, how does the proposed metric fare when the emission path transitions to linear incoherence (where only intensity modulation is viable)?

6. Citations: I suggest referencing the work of Sinefeld, David, et al. titled "Three-photon adaptive optics for mouse brain imaging." (Frontiers in neuroscience 16, 2022: 880859). This paper demonstrates the feasibility of adaptive optics (wavefront shaping) for fluorescence imaging with thin fluorescence samples, which provides intuition and explanation that performing 1-photon fluorescence wavefront shaping may not be challenging for thin brain slice in the author's case.

We have added the citation, though we did not find any mentioning of 1-photon fluorescence wavefront shaping, and we will be happy if the reviewer can give us some more guidance.

7. Optimization Details and Results Discussion: The main text is rather scant on optimization specifics and a thorough discussion of the primary findings. I suggest the authors to add some essential information about these to the main text.

Minor Comments:

1. Figures & Colorbars: Neuronal figures lack colorbars, which makes it appear that the uncorrected images might be saturated. It would be beneficial to include colorbars without normalization to depict the actual values.

2. Uncited Reference: Reference [35] is conspicuously uncited in the manuscript.

3. Terminology Clarification: In the main text, line 245, the term "one-hot vectors" is used. Could the authors clarify if they mean that these vectors are binary?

4. Typographical Error: There's a typo in the main text, line 179. "then" should be corrected to "than". Please carefully proofread the manuscript.

REVIEWER COMMENTS

Reviewer #1 (Remarks to the Author):

I have read the revised paper and the authors' revision comments. I will respond to the authors' response to the major points from my original review.

The first major point here concerns the question of whether the dual SLM correction provides better correction than a single SLM. The authors have presented more experiments to try to answer this question. These results are - even by the admission of the authors at the end of supplementary section 5 - inconclusive. They also state that due to photobleaching arising from the necessary longer exposure, the effectiveness of the overall process is worse.

The second major point concerns the novelty of the "confocal score". In the response the authors write "A mathematical analysis of previously introduced scores which proves why they should favor a focusing modulation, and a proof that a confocal score results in a non-linear function, like a 2-photon score. To our best knowledge no derivation of this form was ever provided in the literature." It is well known from theory of scanning optical microscopes from the 80s and 90s that the image formation in confocal and two photon microscopes is similar (see e.g. the book by Min Gu "Principles of Three-Dimensional Imaging in Confocal Microscopes", 1996) . Therefore, it will be no surprise to those familiar with this theory that the "scores" would be similar for these two instruments. As mentioned in the original review, this is not fundamentally new, as it has formed the basis in effect of all confocal microscope image based AO systems (by choosing a region of pixels on a camera, one is implementing something that is optically equivalent to the pinhole of a confocal microscope). The authors do not need to claim novelty here, and omitting to mention a connection to the previously well-known theory that underpins previous implementations is not helpful to those reading the paper who would benefit from understanding this connection.

As an additional point, I would like to comment on the remarks of Reviewer 2, who wrote "the integral of a two-photon signal in a uniformly stained, infinite volume remains a constant, irrespective of the focus size", to which the authors provided agreement. I think this is a misunderstanding. It is true that if one changes the numerical aperture of the focussing lens, while maintaining the total laser power, then the signal does not change. However, I believe this is not true in the case of introducing phase aberrations, where the total signal does reduce as aberrations increase. This would apply both to confocal and two-photon microscopes.

Reviewer #2 (Remarks to the Author):

I appreciate the authors for addressing my questions and supplying additional data. Again, I support the publication of the work in Nature Communications. Even though the experimental results demonstrated in this work are not necessarily better than those from other existing methods (e.g., variation maximization), the manuscript contains new interesting data and insights that may inspire future development of adaptive optics techniques for deep-tissue imaging.

Reviewer #2 (Remarks on code availability):

The repository is empty!

Reviewer #3 (Remarks to the Author):

All my comments were addressed. No more questions.

Dear reviewers,

We thank you all for your time and for your thoughtful comments. Please find below detailed responses to reviewing comments.

Reviewer #1 (Remarks to the Author):

I have read the revised paper and the authors' revision comments. I will respond to the authors' response to the major points from my original review.

The first major point here concerns the question of whether the dual SLM correction provides better correction than a single SLM. The authors have presented more experiments to try to answer this question. These results are - even by the admission of the authors at the end of supplementary section 5 - inconclusive. They also state that due to photobleaching arising from the necessary longer exposure, the effectiveness of the overall process is worse.

Our original submission did not aim to claim that two modulations are required, and we apologize if such an impression was made. Since this question was raised by two reviewers, we have further investigated it in the revision, and indeed the conclusion is that using the same modulation for both wavelength is a reasonable approximation, since the excitation and emission wavelengths are similar.

It is worth pointing out that the observation that the confocal score can also be optimized using two different modulations in each wavelength can possibly be useful for a confocal correction of a 2P microscope. In 2P the excitation and emission wavelengths are different. One can correct both the illumination and emission arms and achieve a 3rd order non-linearity (equivalent to a 3P excitation). However, we clearly did not demonstrate this in our paper.

The second major point concerns the novelty of the "confocal score". In the response the authors write "A mathematical analysis of previously introduced scores which proves why they should favor a focusing modulation, and a proof that a confocal score results in a non-linear function, like a 2-photon score. To our best knowledge no derivation of this form was ever provided in the literature." It is well known from theory of scanning optical microscopes from the 80s and 90s that the image formation in confocal and two photon microscopes is similar (see e.g. the book by Min Gu "Principles of Three-Dimensional Imaging in Confocal Microscopes", 1996) . Therefore, it will be no surprise to those familiar with this theory that the "scores" would be similar for these two instruments. As mentioned in the original review, this is not fundamentally new, as it has formed the basis in effect of all confocal microscope image based AO systems (by choosing a region of pixels on a camera, one is implementing something that is optically equivalent to the pinhole of a confocal microscope). The authors do not need to claim novelty here, and omitting to mention a connection to the previously well-known theory that underpins

previous implementations is not helpful to those reading the paper who would benefit from understanding this connection.

We agree that emphasizing the equivalence between a 2P microscope and 1P confocal microscope is helpful, and we have revised the manuscript to mention it. We also added a citation of the book. Still the book studies this equivalence in an aberration-free setting. To our best knowledge we are the first to show that the non-linearity of the 2P microscope is equivalent to a 1P confocal microscope also in the presence of aberrations.

As an additional point, I would like to comment on the remarks of Reviewer 2, who wrote “the integral of a two-photon signal in a uniformly stained, infinite volume remains a constant, irrespective of the focus size”, to which the authors provided agreement. I think this is a misunderstanding. It is true that if one changes the numerical aperture of the focusing lens, while maintaining the total laser power, then the signal does not change. However, I believe this is not true in the case of introducing phase aberrations, where the total signal does reduce as aberrations increase. This would apply both to confocal and two-photon microscopes.

Our understanding is that in the presence of an unbounded fluorescent volume, the two-photon signal is invariant to any choice of phase aberration, and not only to a change in the numerical aperture. We attach a sketch of the proof we have derived named “Invariance2P”. (We emphasize that the attached proof is not included in the manuscript and the reviewers should not feel obligated to read it).

However, we believe this discussion exceeds the scope of our current manuscript. Our manuscript does not include a statement about invariance to phase aberration. Rather, we only refer the reader to the analysis of Katz et al, who analyze the invariance to a change in the numerical aperture:

“Noninvasive nonlinear focusing and imaging through strongly scattering turbid layers.”
Ori Katz, Eran Small, Yefeng Guan, and Yaron Silberberg. 2014

Reviewer #2 (Remarks to the Author):

I appreciate the authors for addressing my questions and supplying additional data. Again, I support the publication of the work in Nature Communications. Even though the experimental results demonstrated in this work are not necessarily better than those from other existing methods (e.g., variation maximization), the manuscript contains new interesting data and insights that may inspire future development of adaptive optics techniques for deep-tissue imaging.

Reviewer #2 (Remarks on code availability):

The repository is empty!

We would appreciate it if you could check again. We are not sure when this review was written, but the repository content was uploaded on Jan 02, 2024.

Reviewer #3 (Remarks to the Author):

All my comments were addressed. No more questions.

REVIEWERS' COMMENTS

Reviewer #1 (Remarks to the Author):

The authors have responded to the points I raised in the second review. I am still of the opinion that they could have made further improvements. However, I do not want to stand in the way of this getting published. I just ask that the authors read my points below and make minor adjustments if they wish. I am sure that readers who know the background literature would be able to make their own judgements about whether previous knowledge has been suitably referenced.

The editor may accept the paper if they inclined to do so. Addressing the following points is optional.

Point 1:

I am glad that the authors have provided a reference to the book I suggested. However, I still disagree with the statement in their response: "To our best knowledge we are the first to show that the non-linearity of the 2P microscope is equivalent to a 1P confocal microscope also in the presence of aberrations." A search on Google Scholar for "aberrations confocal two photon microscopy" yields several papers. The first linked paper (<https://onlinelibrary.wiley.com/doi/full/10.1111/j.1365-2818.2011.03544.x>) shows exactly this equivalence (Eq. 8 and the subsequent text). I am sure the same matter is explained in earlier papers too.

Dear reviewers,

We thank you all for your time and for your thoughtful comments. Please find below detailed responses to reviewing comments.

Reviewer #1 (Remarks to the Author):

The authors have responded to the points I raised in the second review. I am still of the opinion that they could have made further improvements. However, I do not want to stand in the way of this getting published. I just ask that the authors read my points below and make minor adjustments if they wish. I am sure that readers who know the background literature would be able to make their own judgements about whether previous knowledge has been suitably referenced.

The editor may accept the paper if they inclined to do so. Addressing the following points is optional.

Point 1:

I am glad that the authors have provided a reference to the book I suggested. However, I still disagree with the statement in their response: "To our best knowledge we are the first to show that the non-linearity of the 2P microscope is equivalent to a 1P confocal microscope also in the presence of aberrations." A search on Google Scholar for "aberrations confocal two photon microscopy" yields several papers. The first linked paper (<https://onlinelibrary.wiley.com/doi/full/10.1111/j.1365-2818.2011.03544.x>) shows exactly this equivalence (Eq. 8 and the subsequent text). I am sure the same matter is explained in earlier papers too.

Thanks for the reference, we added it to the manuscript. We also revised the text of the section deriving the confocal score, and this section now starts upfront by mentioning the prior work establishing equivalence with the 2P. We hope the new wording will not convey an impression that we are over-claiming novelty.